# Secondary Primary Cancer after Primary Gastric Cancer: Literature Review and Big Data Analysis Using the Health Insurance Review and Assessment Service (HIRA) Database of Republic of Korea

**DOI:** 10.3390/cancers14246165

**Published:** 2022-12-14

**Authors:** Jeong Ho Song, Yeonkyoung Lee, Jaesung Heo, Sang-Yong Son, Hoon Hur, Sang-Uk Han

**Affiliations:** 1Department of Surgery, Ajou University School of Medicine, Suwon 16499, Republic of Korea; 2Department of Radiation Oncology, Ajou University School of Medicine, Suwon 16499, Republic of Korea

**Keywords:** gastric cancer, secondary primary cancer, Republic of Korea

## Abstract

**Simple Summary:**

With the development of new chemotherapeutic agents and advances in surgical techniques, the overall survival of gastric cancer patients continues to rise. Long-term follow-up of gastric cancer patients has resulted in an increase in the number of reports of secondary primary cancer (SPC) in these patients. This review analyzes the SPCs of gastric cancer survivors using the nationwide claims data of Republic of Korea, and summarizes previous studies of the incidence of SPC in national and institutional databases.

**Abstract:**

Advances in cancer screening and early detection, as well as improvements in surgical techniques and therapeutics, have contributed to decreasing gastric cancer mortality. The number of gastric cancer survivors continues to rise; however, long-term follow-up has revealed an increase in the risk of post-gastrectomy symptoms or other health problems, such as extra-gastric secondary primary cancer (SPC), in these survivors. Therefore, evidence-based screening for new primary cancer is needed in these populations; however, the incidence of SPC varies by country or continent and its characteristics have not been clearly reported. The characteristics of SPC are of increasing interest to both treatment providers and gastric cancer survivors; thus, this literature review explores not only the epidemiology and biology of SPC but also clinical and biological factors that influence its prognosis.

## 1. Introduction

Cancer research and advances in cancer treatment modalities have increased the rates of overall survival in cancer patients. After completion of cancer treatment, cancer survivors return to pre-treatment life, but their quality of life is not the same as before treatment. Patients face financial burden, psychological difficulty, work-related problems, and physiological changes in the body after organ resection [1]. Along with providing emotional support and explaining the disease to patients, health care providers should also be aware that secondary primary cancers (SPCs) can develop in cancer survivors.

In gastric cancer, the development of new chemotherapeutic agents and advances in surgical techniques have resulted in more survivors in recent decades [2,3,4,5,6]. In East Asian countries, the advent of national screening programs has resulted in more than half of all gastric cancer patients being diagnosed with early stage tumors, and their 5-year survival rate exceeds 90% [2,7,8]. SPCs affect the long-term survival of cancer survivors, and several national and institutional database studies have been published previously. Therefore, the purpose of this study is to review the literature describing the occurrence of SPCs in gastric cancer patient survivors and to analyze the SPCs of Korean gastric cancer survivors using a Korean nationwide claims database.

## 2. Epidemiology of SPCs

Previous studies on the incidence of SPCs can be divided into two categories according to their scope, namely single institution- and nationwide population-based studies. Single institution-based studies with more than one thousand patients are summarized Table 1 [9,10,11,12,13,14,15,16,17,18,19]. Most studies on SPCs were performed in regions with a high prevalence of gastric cancer. The incidence of SPC in gastric cancer patient survivors varies between 1.0% and 6.6%. Synchronous tumors were defined as those with a time interval between SPC and gastric cancer diagnosis of less than 6 months, and metachronous tumors were defined as those with more than 6 months. The development of SPCs after gastric cancer mainly occurred within 2 years of diagnosis including synchronous tumor [15,17,18].

Nationwide population-based studies in Japan, Taiwan, United States of America, Portugal, and Sweden are available and listed in Table 2 [20,21,22,23,24]. The incidence of SPCs in patients with gastric cancer ranges from 4.4% to 5.5%. In most studies, the risk of SPC in patients with gastric cancer was estimated using standardized incidence ratio (SIR), which is defined as the number of observed cancer occurrences divided by the expected number. The expected number of cancers was calculated by multiplying the first primary cancer incidence by the person-years after first primary cancer diagnosis in the general population. The SIR for all SPCs in gastric cancer patients ranges from 0.91 to 1.46. The most common SPC site after gastric cancer is the gastrointestinal tract, such as esophagus, small intestine, and colon. Some studies showed increased incidences of SPCs in thyroid, pancreas, bone and soft tissues, ovaries, bladder, kidney, liver, and intrahepatic bile duct, and non-Hodgkin’s lymphoma. By contrast, the incidence of SPCs in lung, breast and prostate, melanoma, and myeloma decreased after a diagnosis of gastric cancer.

In both men and women, the incidences of SPCs in the gastrointestinal tract and non-Hodgkin’s lymphoma after diagnosis of gastric cancer were high. In men, the incidence of SPCs were elevated in prostate and bladder, whereas in women, the incidences of SPCs were elevated in bone and soft tissue, ovaries, kidneys, liver and intrahepatic bile ducts. The incidence of SPCs in gastric cancer survivors was higher in men than in women [12,19,21,23].

## 3. Biology of SPCs

Several risk factors for gastric cancer have been documented, namely *Helicobacter pylori* infection, male sex, old age, low socioeconomic status, smoking, alcohol consumption, salted food intake, preserved food, genetic factors, and familial predisposition [25]. These factors often overlap with risk factors for other cancers.

Consumption of preserved foods increases the risk of gastric cancer, hepatocellular carcinoma, nasopharyngeal carcinoma breast cancer, and prostate cancer [26,27,28,29,30,31]. By contrast, low intake of vegetables is known to lead to gastric cancer, lung cancer, hepatocellular carcinoma, and nasopharyngeal cancer [25,32,33,34]. Smoking is the major risk factor for cancers at several sites, such as stomach, oropharynx, larynx, esophagus, and lung [25,35,36,37,38,39]. In a previous study, seven out of nine patients (77.7%) with SPCs after gastric cancer consumed tobacco [40]. A synergy between smoking and alcohol consumption was also identified as contributing to carcinogenesis [41,42].

Chemo/radiotherapy has been proposed to contribute to the development of SPCs. Treatments for the first malignancy may contribute to the development of SPCs by causing damage to specific regions of DNA, chromosomal rearrangements, or chromosome loss [43,44]. Some genetic factors have been found to be associated with the development of gastric and other cancers. Microsatellite instability is closely associated with gastric and sporadic colorectal carcinoma [45,46,47]. Lynch syndrome, an autosomal dominant inherited cancer predisposition syndrome, is a risk factor for tumors of gastric, colorectal, small bowel, gallbladder, and biliary tract, and is also associated with microsatellite instability [48,49,50]. In addition, several hereditary diseases, such as Li-Fraumeni syndrome, breast-ovarian cancer syndrome, multiple endocrine neoplasia, Peutz-Jegher syndrome, and Cowden syndrome, increase the risk of gastric and other cancers [40,51].

## 4. Predisposing Factors for SPCs

A Taiwanese population-based study employing multivariate logistic regression analysis demonstrated that age over 70 years, male sex, and comorbidities, such as diabetes mellitus, chronic obstructive pulmonary disease, and liver cirrhosis are risk factors for developing SPCs [21]. Several single institution-based studies found several risk factors for developing SPCs after gastric cancer, such as age over 60 years, differentiated histology, earlier stage gastric cancer, and multiplicity of lesions [14,19].

## 5. Prognosis of Patients with SPCs

The long-term outcomes of gastric cancer patients with SPCs have not been reported by national population-based studies. However, several single-institutional retrospective studies have reported the long-term survival outcomes of gastric cancer patients with SPCs. In a Korean study, the 5-year overall survival rate according to gastric cancer stage regardless of SPC site was as follows; 61% for stage I, 39% for stage II, 30% for stage III, and 0% for stage IV [16]. In studies comparing the survival outcomes of patients with and without SPCs, the 5-year survival rates of patients with SPCs were statistically poorer than those of patients without SPCs [18,19]. In addition, the 5-year survival rate of patients without SPCs was 76.5%, that of patients with metachronous SPC was 67.5%, and that of patients with synchronous SPC was 34.1%, which were statistically significant (*p* < 0.001 for metachronous SPC vs. no SPC and *p* < 0.001 for synchronous SPC vs. no SPC). In a Japanese study analyzing 10-year survival rates, the survival rate of patients with metachronous SPC was higher than that of patients without SPC or with synchronous SPC (75.2% for metachronous SPC; 69.3% for no SPC; and 40.1% for synchronous; *p* < 0.01) [11]. The 5-year survival of metachronous SPC patients was significantly superior to that of synchronous SPC patients in a Korean retrospective study (77.9% for metachronous SPC and 55.2% for synchronous SPC; *p* = 0.002) [17]. However, in a Spanish study, no significant differences in survival rates were found between patients with synchronous and those with metachronous SPC [15].

## 6. SPCs after Gastric Cancer in Republic of Korea

### 6.1. Materials and Methods

Nationwide population and cancer data were obtained from the public medical insurance system of Republic of Korea, the National Health Insurance (NHI) system. The Health Insurance Review & Assessment (HIRA) database is an involuntary government-operated organization that builds accurate review and quality assessments for the NHI system. The number of people registered in the NHI was 52,272,755 in 2016, representing about 97% of the total population of Republic of Korea. The HIRA database provides information on age, sex, diagnostic codes, prescribed drugs, and treatment, including surgical history and procedures.

The nationwide cohort was abstracted from the HIRA claims data and included patients who were diagnosed with gastric cancer from January 2009 to December 2016. Firstly, patients with gastric cancer (C16) were confirmed by the Korean Classification of Disease, sixth edition (KCD-6), a modified version of the International Classification of Disease 10 (ICD-10) for the Korean health care system. SPC was defined as another ‘first’ diagnosis of primary cancer with an interval of 6 months or more after diagnosis of gastric cancer in the HIRA dataset. Patients already diagnosed with other cancers prior to gastric cancer were excluded from the study. This study was approved by the institutional review board of Ajou University Hospital (AJIRB-MED-EXP-17-101).

Descriptive statistics were calculated to estimate the frequency of SPC using the diagnostic code assigned on the first hospital visit. Additionally, SIRs and the corresponding 95% confidence interval (CI) were analyzed to estimate the relative risk of SPC in patients with gastric cancer, as compared with the general population. The CI was calculated using normal approximation, and the mid-*p* exact test was used when the observed value was below 5. All statistical analyses were performed using R-statistics, version 3.0.2.

### 6.2. Results

During the follow-up period between 2009 and 2016, we identified 225,973 patients with gastric cancers who developed 10,324 SPCs (4.7%). The incidence of SPCs tended to increase with age, and was the highest in the 70–79 years old age group (6.0%, about one-third of all SPCs), followed by the 60–69 (5.6%), 50–59 (3.6%), 40–49 (3.1%), 30–39 (2.7%), and 20–29 (2.5%) age groups (Table 3).

The median diagnostic age was 66 years and the time from primary gastric cancer to an SPC varied from 8.0 to 43.2 months according to the type of SPC, with a median of 27.1 months; cancers of tongue, oral cavity, or anus were diagnosed early within 1 year while those of ureter, bladder, and brain were diagnosed relatively late (30 months or later) (Table 4). The most common type of SPC in gastric cancer patients was colorectal cancer, which represented 15.7% of all SPCs, followed by lung (15.3%), liver (10.1%), prostate (8.6%), and thyroid (6.9%). The frequencies of SPCs according to sex are shown in Figure 1. The overall incidences of SPCs in male and female patients with primary gastric cancer were 4.9% and 3.8%, respectively. In males, the five most common types of SPCs were lung cancer (17.9%), colorectal cancer (16.0%), prostate cancer (11.7%), liver cancer (11.6%), and esophageal cancer (5.0%). In females, they were thyroid cancer (15.6%), colorectal cancer (14.8%), breast cancer (10.0%), lung cancer (8.3%), and liver cancer (5.9%).

People with primary gastric cancer had an elevated risk of developing an SPC. The overall SIR for any cancer was 1.35 (1.38 in males and 1.13 in females) (Table 5). In males, SIRs over 2.0 were observed for breast (5.09), esophagus (3.64), bone/cartilage (3.10), anus (2.79), small intestine (2.67), ureter (2.60), tongue (2.54), bladder (2.28), pancreas (2.23), oral cavity (2.19), and gallbladder/bile duct (2.07) cancers. The SIR was lower than 1.0 for thyroid (0.87) cancer. In females, the highest SIR was observed for esophagus (11.17) cancer, followed by tongue (9.24), anus (5.49), bone/cartilage (3.47), oral cavity (3.41), and small intestine (3.32) cancers, as well as for patients with immunoproliferative disease (3.71) and leukemia (3.18). SIRs for thyroid, breast, cervix, or uterus were lower than 1.0. Regarding the time interval between the primary gastric cancer and the SPC, the incidence risk of SPC was highest within 1 year (SIR 2.42), and 1.56 at 1–2 years (Figure 2).

### 6.3. Discussion

This study includes the largest data set ever reported for evaluating the national incidence of SPCs among gastric cancer survivors using claims data from the HIRA database in Republic of Korea. The major strengths of this study are the large sample size (10,324 SPCs among 225,973 gastric cancer patients) and the accuracy of the reported data, which can be attributed to the mandatory requirement for cancer registration in the health insurance system of Korea. Therefore, we suggest that the results may be more reliable than those of previous studies.

Several previous studies of SPCs in gastric cancer survivors have revealed that they share several common features. The incidence of SPCs in gastric cancer survivors is clearly higher than that in the general population. Due to the nature of the therapies used to treat primary cancers patients, the genetic predispositions of these patients, as well as the common environmental factors that lead to cancer, cancer survivors are at a higher risk of developing SPCs than the general population [1]. In addition, patients with malignancies tend to be more health-conscious or have greater access to hospitals than the general population. SPC increases with age and occurs most often within the 2-year period following a gastric cancer diagnosis. Although the overall incidence increases with age, the SIR for SPC is the highest in relatively young gastric cancer patients [21,52,53]. Young cancer patients are considered to be genetically predisposed to cancer, and careful follow-up and screening strategies are required. Furthermore, SPCs in gastric cancer survivors develop more frequently in males than in females [11,14,16,23].

The most common types of SPCs in gastric cancer patients are tumors of the digestive tract, such as those of the esophagus, small bowel, and colon. The pathophysiology underlying the high incidence of digestive organ cancers is unclear but may be due to factors affecting the gastrointestinal tract, such as N-nitroso compounds, drinking alcohol, and smoking, as well as common carcinogenic processes [54,55,56]. Another possibility for the high incidence is that these cancers are readly detected to regular gastric cancer follow-up examinations, such as endoscopy and abdominopelvic computed tomography [20,57,58]. The high incidence of intra-abdominal SPCs, such as hepatocellular carcinoma, pancreas cancer, gallbladder cancer, and bile duct cancer, in gastric cancer patients is also thought to be due to regular follow-up examinations for recurrence after gastric cancer treatments.

The exact mechanism involved in the development of SPC after gastric cancer has not been fully elucidated. Genetic vulnerability, immunologic factors, and exposure to carcinogens including those used in gastric cancer treatments, as well as common risk factors between gastric cancer and SPC, are considered important [13,44]. Further studies are needed to verify the causal relationship between gastric cancer and SPCs.

Few studies have analyzed survival outcomes according SPC type because SPCs are rare. Survival analysis performed irrespective of SPC type is the main method. However, each type of cancer has a different prognosis. The prognosis of patients with early stage gastric cancer is determined by the types of SPC they acquire, not by the gastric cancer itself. To analyze the true prognosis of gastric cancer patients with an SPC, detailed analysis according to SPC type and stages of gastric cancer and SPC are needed.

This study has several limitations. First, the HIRA dataset does not include histologic types or germline mutations in mismatch repair genes of primary gastric cancer or SPCs; therefore, we were unable to evaluate their relationships with various cancers (e.g., Lynch syndrome). Additionally, we were unable to elucidate the causal relationship and mechanisms between gastric cancer treatment (surgery, endoscopic resection, and chemotherapy) and SPC type. Third, detailed individual data and cancer stages were not considered in this study. This is a common limitation of studies using the HIRA dataset. Therefore, we were unable to evaluate whether there was difference in the incidence of SPC according to the stage of gastric cancer. However, based on the observation that about 88% of SPCs are detected within 5 years after a gastric cancer diagnosis, this study suggests that follow-up protocols for the early detection of SPCs could improve the survival of gastric cancer patients.

## 7. Conclusions

This study showed that gastric cancer survivors had an increased likelihood of being identified with SPC. The occurrence of SPCs after gastric cancer treatment increased with age and was common in the digestive tract. Close surveillance after gastric cancer treatment should be considered during follow-up not only to detect recurrence but also to ensure the early detection of SPCs. Further research will be needed to elucidate the mechanisms of SPC development and the prognosis of patients with SPCs.

## Figures and Tables

**Figure 1 cancers-14-06165-f001:**
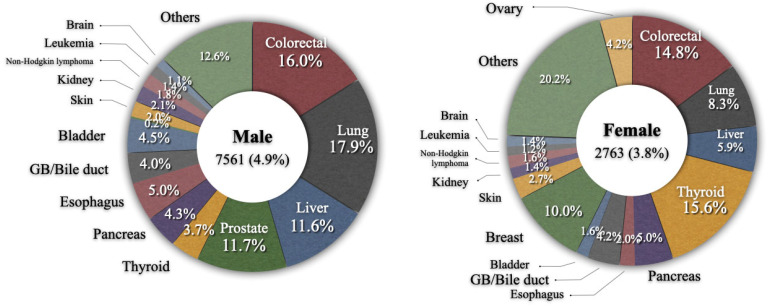
Frequency of secondary primary cancer according to sex.

**Figure 2 cancers-14-06165-f002:**
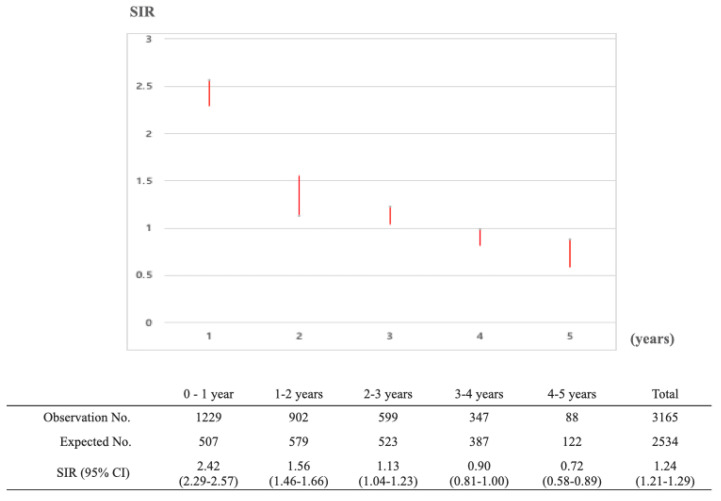
Standardized incidence ratios of secondary primary cancer along the follow up periods in patients who were previously diagnosed with primary gastric cancer between January 2010 and December 2014.

**Table 1 cancers-14-06165-t001:** List of single institutional-based studies on the incidence of secondary primary cancer (SPC) after gastric cancer.

Authors	Country	Publication Year	Number of Patients	Incidence of SPC	Synchronous/ Metachronous	Common SPCs
Takekuni K et al. [9]	Japan	1999	1925	127 (6.6%)	N/A	Lung, liver, colorectum, esophagus, and pancreas
Dinis-Ribeiro M et al. [10]	Portugal	2002	2268	78 (3.4%)	21/57	Colorectum, breast, uterus, urinary tract, and lymphoma
Ikeda Y et al. [11]	Japan	2003	2250	95 (4.2%)	48/47	Colorectum, lung, liver, esophagus, and breast
Ikeda Y et al. [12]	Japan	2005	1070	54 (5.0%)	0/54	Lung, colorectum, esophagus, breast, and liver
Park YK et al. [13]	Korea	2005	2509	65 (2.6%)	17/48	Colorectum, breast, hepatobiliary, esophagus, and uterus
Lee JH et al. [14]	Korea	2006	3291	111 (3.4%)	111/0	Colorectum, lung, esophagus, liver, and breast
Molinero M et al. [15]	Spain	2006	1170	23 (1.96%)	11/12	Colorectum, multiple myeloma, prostate, lung, and kidney
Ha TK et al. [16]	Korea	2007	10,090	96 (1.0%)	96/0	Colorectum, liver, kidney, pancreas and gallbladder
Eom BW et al. [17]	Korea	2008	4593	159 (3.4%)	49/110	Colorectum, lung, liver, kidney, and lymphoma
Kim JY et al. [18]	Korea	2012	5778	214 (3.70%)	0/214	Colorectum, lung, liver, ovary, and cervix
Kim C et al. [19]	Korea	2013	3066	70 (2.5%)	32/38	Colorectum, lung, liver, gallbladder, and head and neck

SPC, secondary primary cancer; N/A, not applicable.

**Table 2 cancers-14-06165-t002:** List of nationwide population-based studies on the incidence of secondary primary cancer (SPC) after gastric cancer.

Country of Study	Publication Year	Number of Patients	Statistical Significance	SPC Site
Japan [20]	2014	174,477	3225 (1.8%) patients had SPC.	SIR for all cancers, 1.09. SIR for esophagus cancer, 1.70 SIR for colorectum cancer, 1.35
Taiwan [21]	2016	47,729	2110 (4.42%) patients had SPC. SIR for all cancer, 1.46	SIR for head and neck, 1.34. SIR for esophagus, 2.16. SIR for colon and rectum, 1.37. SIR for bones and soft tissues, 1.95. SIR for ovaries, 2.89. SIR for bladder, 1.47. SIR for kidneys, 1.44. SIR for non-Hodgkin’s lymphoma, 5.56.
the United States [22]	2016	33,720	1838 (5.45%) patients had SPC with O/E ratio of 1.11.	Gastrointestinal O/E ratio, 1.71. Biliary O/E ratio, 1.30, Pancreatic O/E ratio, 1.60. Thyroid O/E ratio 2.00.
Portugal [23]	2017	7427	331 (4.5%) patients had SPC (26.9% synchronous and 73.1% metachronous). 10-year cumulative incidence of SPC, 4.8% SIR for all cancer, 1.30 (male) and 1.20 (female)	SIR for esophagus, 4.99 (male) and 8.03 (female). SIR for small intestine, 11.04 (male) and 13.08 (female). SIR for colon, 2.42 (male) and 2.58 (female). SIR for non-Hodgkin’s lymphoma, 2.53 (male). SIR for liver and intrahepatic bile duct, 5.18 (female).
Sweden [24]	2021	23,137	1042 (4.5%) patients had SPC.	SIR for esophagus, 2.15. SIR for small intestine, 4.12. SIR for kidney, 1.62.

SPC, secondary primary cancer; O/E, observed/expected; SIR, Standardized incidence ratio.

**Table 3 cancers-14-06165-t003:** Incidence of secondary primary cancer in patients who were previously diagnosed with primary gastric cancer according to age group.

Age (Years)	Primary Gastric Cancer	Secondary Primary Cancer
20–29	1369	35 (2.5%)
30–39	7915	214 (2.7%)
40–49	26,880	825 (3.1%)
50–59	53,611	1920 (3.6%)
60–69	61,402	3419 (5.6%)
70–79	54,575	3290 (6.0%)
>80	20,221	621 (3.1%)
Total	225,973	10,324 (4.7%)

**Table 4 cancers-14-06165-t004:** Median ages and interval times to diagnosis according to various types of secondary primary cancer.

Type of SPC	Number (Proportion)	Age (Year)	Interval to Diagnosis (Months)
Colorectal	1619 (15.7%)	67	26.5
Lung	1584 (15.3%)	68	31.5
Liver	1043 (10.1%)	66	25.9
Prostate	888 (8.6%)	69	29.9
Thyroid	710 (6.9%)	57	24.8
Pancreas	461 (4.5%)	68	29.9
Esophagus	434 (4.2%)	67	24.5
Gallbladder/bile duct	415 (4.0%)	69	31.4
Bladder	382 (3.7%)	69	32.6
Breast	294 (2.8%)	53	25.3
Skin	224 (2.2%)	71.5	35.7
Kidney	200 (1.9%)	65	23.1
Non-Hodgkin lymphoma	178 (1.7%)	67	29.3
Leukemia	137 (1.3%)	67	28.8
Brain	125 (1.2%)	64	32.9
Ovary	115 (1.1%)	46	22.4
Tongue	100 (1.0%)	61	8.0
Larynx	81 (0.8%)	66	27.5
Small intestine	80 (0.8%)	66	19.8
Immunoproliferative disease	75 (0.7%)	59	26.9
Soft tissue	63 (0.6%)	65	27.6
Bone/Cartilage	61 (0.6%)	60	26.8
Multiple myeloma	60 (0.6%)	68	34.7
Oral cavity	56 (0.5%)	67	9.9
Ureter	50 (0.5%)	65.5	43.2
Cervix	42 (0.4%)	55.5	25.2
Uterus	38 (0.4%)	59	28.5
Melanoma	36 (0.3%)	67.5	29.7
Anus	34 (0.3%)	59.5	8.4
Others	739 (7.2%)	-	-
Overall	10,324	66	27.1

SPC, secondary primary cancer.

**Table 5 cancers-14-06165-t005:** Standardized incidence ratios of secondary primary cancer according to sex.

Type of SPC	Overall				Male (*n* = 153,825)			Female (*n* = 72,148)		
	<5 yrs.	>5 yrs.	SIRs	95%CI	*n* (%)	SIRs	95% CI	*n* (%)	SIRs	95% CI
Colorectal	1450 (89.6%)	169 (10.4%)	1.35	1.33–1.38	1209 (16.0%)	1.49	1.41–1.58	410 (14.8%)	1.66	1.51–1.83
Lung	1347(85.0%)	237(15.0%)	1.64	1.56–1.72	1355 (17.9%)	1.73	1.63–1.82	229 (8.3%)	1.44	1.26–1.64
Liver	935 (89.6%)	108 (10.4%)	1.92	1.83–2.02	880 (11.6%)	1.51	1.41–1.61	163 (5.9%)	1.75	1.49–2.04
Prostate	750 (84.5%)	138 (15.5%)	-	-	888 (11.7%)	1.92	1.80–2.05	-	-	-
Thyroid	651 (91.7%)	59 (8.3%)	0.56	0.52–0.61	278 (3.7%)	0.87	0.77–0.97	432 (15.6%)	0.66	0.60–0.73
Pancreas	394 (85.5%)	67 (14.5%)	2.32	2.11–2.54	324 (4.3%)	2.23	1.99–2.48	137 (5.0%)	2.32	1.95–2.74
Esophagus	399 (91.9%)	35 (8.1%)	5.14	4.66–5.64	380 (5.0%)	3.64	3.28–4.02	54 (2.0%)	11.17	8.39–14.58
Gallbladder/bile duct	350 (84.3%)	65 (15.7%)	2.01	1.82–2.22	300 (4.0%)	2.07	1.85–2.32	115 (4.2%)	1.80	1.49–2.16
Bladder	328 (85.9%)	54 (14.1%)	2.81	2.53–3.10	339 (4.5%)	2.28	2.04–2.53	43 (1.6%)	2.49	1.80–3.36
Breast	256 (87.1%)	38 (12.9%)	0.47	0.42–0.53	18 (0.2%)	5.09	3.01–8.04	276 (10.0%)	0.68	0.60–0.76
Skin	173 (77.2%)	51 (22.8%)	1.50	1.31–1.71	149 (2.0%)	1.72	1.45–2.02	75 (2.7%)	1.37	1.07–1.71
Kidney	183 (91.5%)	17 (8.5%)	1.31	1.14–1.51	160 (2.1%)	1.12	0.96–1.31	40 (1.5%)	1.32	0.94–1.79
Non-Hodgkin lymphoma	159 (89.3%)	19 (10.7%)	1.27	1.09–1.47	134 (1.8%)	1.13	0.94–1.33	44 (1.6%)	1.17	0.85–1.56
Leukemia	121 (88.3%)	16 (11.7%)	1.40	1.18–1.66	104 (1.4%)	1.37	1.12–1.66	33 (1.2%)	3.18	2.19–4.46
Brain	108 (86.4%)	17 (13.6%)	1.94	1.62–2.31	87 (1.2%)	1.85	1.49–2.29	38 (1.4%)	1.98	1.40–2.72
Ovary	106 (92.2%)	9 (7.8%)	-	-	-	-	-	115 (4.2%)	2.24	1.85–2.68
Tongue	96 (96.0%)	4 (4.0%)	4.34	3.53–5.28	52 (0.7%)	2.54	1.90–3.33	48 (1.7%)	9.24	6.81–12.25
Larynx	72 (88.9%)	9 (11.1%)	1.94	1.54–2.41	78 (1.0%)	1.47	1.16–1.83	3 (0.1%)	1.81	0.36–5.28
Small intestine	72 (90.0%)	8 (10.0%)	3..05	2.42–3.80	57 (0.8%)	2.67	2.02–3.45	23 (0.8%)	3.32	2.10–4.98
Immunoproliferative disease	71 (94.7%)	4 (5.3%)	2.34	1.84–2.93	34 (0.5%)	1.44	1.00–2.02	41 (1.5%)	3.71	2.67–5.04
Soft tissue	56 (88.9%)	7 (11.1%)	1.73	1.33–2.22	46 (0.6%)	1.62	1.18–2.16	17 (0.6%)	1.68	0.98–2.69
Bone/Cartilage	54 (88.5%)	7 (11.5%)	3.29	2.52–4.23	42 (0.6%)	3.10	2.24–4.19	19 (0.7%)	3.47	2.09–5.41
Multiple myeloma	47 (78.3%)	13 (21.7%)	1.30	0.99–1.68	46 (0.6%)	1.37	1.00–1.82	14 (0.5%)	1.04	0.57–1.75
Oral cavity	47 (83.9%)	9 (16.1%)	2.69	2.03–3.50	39 (0.5%)	2.19	1.55–2.99	17 (0.6%)	3.41	1.99–5.47
Ureter	40 (80.0%)	10 (20.0%)	2.91	2.16–3.84	38 (0.5%)	2.60	1.84–3.57	12 (0.4%)	2.92	1.51–5.10
Cervix	37 (88.1%)	5 (11.9%)	-	-	-	-	-	42 (1.5%)	0.49	0.36–0.67
Uterus	33 (86.8%)	5 (13.2%)	-	-	-	-	-	38 (1.4%)	0.77	0.55–1.06
Melanoma	29 (80.6%)	7 (19.4%)	1.92	1.34–2.66	19 (0.3%)	1.58	0.95–2.47	17 (0.6%)	2.65	1.54–4.24
Anus	32 (94.1%)	2 (5.9%)	3.67	2.54–5.13	15 (0.2%)	2.79	1.56–4.60	19 (0.7%)	5.49	3.30–8.57
Others	680 (94.1%)	59 (5.9%)	-	-	490 (6.5%)	-	-	249 (9.0%)	-	-
Overall	9076 (87.9%)	1248 (12.1%)	1.35	1.33–1.38	7561 (4.9%)	1.38	1.33–1.41	2763 (3.8%)	1.13	1.09–1.18

SPC, secondary primary cancer; SIRs, standardized incidence ratios; CI, confidence interval.

## Data Availability

The data used in the study are potentially identifiable and are not publicly available. The raw claims datasets generated and/or analyzed during the study are not publicly available due ethical restrictions by the HIRA.

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
