# Peer review of "Secondary Primary Cancer after Primary Gastric Cancer: Literature Review and Big Data Analysis Using the Health Insurance Review and Assessment Service (HIRA) Database of Republic of Korea"

_cancers, 2022, doi:10.3390/cancers14246165_

Round 1

Reviewer 1 Report

Dear Authors,

congratulations for the quality of your paper and the good organization behind your study. To accomplish my role, I would like to ask a couple of question:

- In which way is the KCD-6 a modified version of ICD-10?

- In your discussion, paragraph you describe the limitations of the HIRA dataset regarding the lack of histologic and genetic assessment: have you considered an underestimation of hereditary and familiar syndromes? Is there any possibility to integrate the datas coming from prevalence and incidence of FAP Syndrome, MCRAs, MAP, Lynch Syndrome and others with your description? Because in this case it will change your conclusion: the key is not in the follow-up, but in a proper surveillance according to a prior genetic assessment. 

Despite those questions I found your descriptive analysis well done and I'll be happy to receive your answers.

Best Regards

Author Response

Responses to the comments from Reviewer #1

The comments from Reviewer #1 were as follows.

Dear Authors,

congratulations for the quality of your paper and the good organization behind your study. To accomplish my role, I would like to ask a couple of question:

  • We appreciate your positive comments and valuable reviews on our manuscript.

- In which way is the KCD-6 a modified version of ICD-10?

  • The Korean Standard Classification of Diseases (KCD) is a modification of the International Classification of Diseases (ICD) announced by the World Health Organization for the Korean health care system. KCD is essentially the same as ICD in oncological parts, but it is reflected a little later after ICD distribution. 

- In your discussion, paragraph you describe the limitations of the HIRA dataset regarding the lack of histologic and genetic assessment: have you considered an underestimation of hereditary and familiar syndromes? Is there any possibility to integrate the datas coming from prevalence and incidence of FAP Syndrome, MCRAs, MAP, Lynch Syndrome and others with your description? Because in this case it will change your conclusion: the key is not in the follow-up, but in a proper surveillance according to a prior genetic assessment.

  • As you pointed out, there may be some data where a second primary cancer integrated in a patient with familial gastric cancer. However, the incidence of second primary cancer following each syndrome is extremely lower and cannot be analyzed in this study through HIRA data. Further research is needed on the effectiveness of proper surveillance following prior genetic assessment you mentioned.

Reviewer 2 Report

This is a very well written review emphasizing  the risk of secondary primary cancer from gastric origin and its wide range of locations. Even if we do not have physiopathological explanations, this very important work of synthesis is very educative, and enforcing the need for close follow up.

No additional comment

Author Response

Thank you very much for your positive comments.

Reviewer 3 Report

Thank you for the opportunity to review this manuscript. The topic of second primary malignancy or cancers seems to be very important since anticancer therapy of first primary cancer can contribute to development of SPC. The Korean results are well organised and the set of patients is very large! In my opinion presented review brings important data about SPC among gastric cancer population in Korea, and this review should be published.

Author Response

(The authors gave the same response as above.)
